# Development of Combretastatin A-4 Analogues as Potential Anticancer Agents with Improved Aqueous Solubility

**DOI:** 10.3390/molecules28041717

**Published:** 2023-02-10

**Authors:** Zhi-Hao Chen, Run-Mei Xu, Guang-Hao Zheng, Ye-Zhi Jin, Yuan Li, Xin-Yuan Chen, Yu-Shun Tian

**Affiliations:** Key Laboratory of Natural Medicines of the Changbai Mountain, Ministry of Education, College of Pharmacy, Yanbian University, Yanji 133002, China

**Keywords:** CA-4 analogs, cyano, structural development, anticancer, aqueous solubility

## Abstract

Combretastatin A-4 (CA-4) is a potent tubulin polymerisation inhibitor. However, the clinical application of CA-4 is limited owing to its low aqueous solubility and the easy conversion of the olefin double bond from the more active *cis-* to the less active *trans*-configuration. Several structural modifications were investigated to improve the solubility of CA-4 derivatives. Among the compounds we synthesized, the kinetic solubility assay revealed that the solubility of compounds containing a piperazine ring increased the most, and the solubility of compounds 12a1, 12a2, 15 and 18 was increased 230–2494 times compared with that of the control compound (*Z*)-3-(4-aminophenyl)-2-(3,4,5-trimethoxyphenyl)acrylonitrile (**9a**). In addition, these synthesised stilbene nitriles had high anticancer cell (AGS, BEL-7402, MCF-7, and HCT-116) selectivity over L-02 and MCF-10A normal cells while maintaining micromolar activity against cancer cells. The most cytotoxic compound is **9a**, and the IC50 value is 20 nM against HCT-116 cancer cells. Preliminary studies indicated that compound **12a_1_** had excellent plasma stability and moderate binding to rat plasma proteins, suggesting it is a promising lead compound for the development of an anticancer agent.

## 1. Introduction

Tubulin participates in the mitotic activity of the cell cycle and plays a key role in eukaryotic cells. If tubulin is interfered with or destroyed by external factors, cancer cells cannot complete mitosis and enter apoptosis [1]. Combretastatins were first isolated from the bark of *Combretum caffrum* [2,3]. One of the structures with outstanding activity is combretastatin A-4 (Figure 1, CA-4), which strongly inhibits tubulin assembly by interacting at the colchicine-binding site [4]. However, CA-4 is a poorly water-soluble and unstable compound [5,6]. Low solubility of a compound will have some negative effects in terms of drug development, such as low bioavailability and a short half-life in vivo [7], and the in vivo anti-tumour effect of the compound may be different from that in vitro [8]. Therefore, in the early stages of the development of new drugs, medicinal chemists need to consider the aqueous solubility in the process of finding lead compounds. Generally, the water solubility of compounds can be improved by chemical structure modification, making compounds into water-soluble prodrugs or formulations methods. However, compared with the risk of increased potential toxicity brought on by the formulation, it may be better to improve the water solubility of compounds by chemical modification or making compounds into water-soluble prodrugs.

Efforts to overcome the poor solubility of CA-4 have led to some compounds with excellent water-solubility, such as the disodium phosphate derivative of CA-4 (Figure 1, CA-4P) and AVE8062 (Ombrabulin, a water-soluble amino acid salt prodrug of CA-4 analogue) [9,10], both of them had successfully entered clinical trials and had performed well in early clinical trials. In the phase I clinical trials, CA-4P was injected intravenously as a single reagent, which can rapidly dephosphorize to CA-4, and a reduction in tumor blood flow was observed in most patients [11,12,13]. This is consistent with the effect of CA-4P as a vascular disrupting agent (VDA) in preclinical trials [14]. This vascular disrupting activity causes the tumor tissue vascular collapse, cutting down the oxygen and nutrient supply to tumor cells, leading to tumor cell death. Similarly, colchicine and the vinca alkaloids that also target tubulin showed vascular targeting activity. However, unlike the latter, the clinical data of CA-4P show that its half-life is short (about 30 min) and can be quickly removed from the blood, this led to short-term tissue exposure, which might explain its action doses being well below its MTD, and show great anti-cancer potential and therapeutic effect [11,12,13]. The subsequent phase II clinical trial studied its therapeutic effect on anaplastic thyroid cancer (ATC), but this study could not prove the median improvement in survival rate and disease response. Therefore, the study was terminated in 2007 [15]. In terms of combined treatment with other drugs, the phase III clinical trial for ATC patients conducted by Oxigene in 2007 showed that CA-4P effectively improved the overall survival rate of patients and had an acceptable safety profile. Still, the goal of doubling the median survival rate has not been achieved [16]. In addition, the early research reports on CA-4 pointed out that the *Z*-restricted configuration can quickly convert into 100-fold less active *trans* isomerization under the conditions of absorbing heat, light and protonic media (configuration instability) [6]. Up to now, *cis*-stable CA-4 analogues that have not been tested in clinical trials, and compounds such as CA-4P and AVE8062 that have entered clinical trials contain isomeric olefinic bonds, which may become one of the factors hindering the sustained clinical success of these compounds.

We have previously reported the anticancer efficacy of the CA-4 derivative **2** (Figure 1) [17]. This compound has a cyano group introduced into the ethylene bond of the stilbene structure; due to the cyano-steric effect, the main body of the compound was fixed to a rigid structure and proved to be a *Z* isomer by nuclear overhauser effect (NOE), which solved the problem of the instability of these types of compounds (compound **2** is configuration stable), and had an IC_50_ value against cancer cells at the micromolar level. However, preliminary pharmacological activity testing indicated that the aqueous solubility of compound **2** (0.40 mg/100 mL) was poorer than that of CA-4 (5.77 mg/100 mL). We speculated that the introduction of the cyano group on the olefin double bond made the whole structure of the stilbene nitrile more rigid, which decreased the solubility in water. The cyano group on the olefinic double bond was essential to maintain the *cis* configuration of the parent stilbene [17,18]; therefore, we attempted to improve the aqueous solubility of the stilbene via modification at other positions of the core structure. Previous structure–activity relationship studies have indicated that the methyl group at the *para* position of the phenyl B ring was the key to its activity [17] (compound **2**, Figure 1). Still, the methyl group is difficult to modify chemically. Therefore, we searched for a highly active compound that could be easily modified while retaining the parent structure. Finally, we found **9a** (Figure 1), which had higher anticancer activity against AGS, HCT-116 and BEL-7402 cell lines, compared with **2**. Because of the nature of the parent structure, **9a** (two-fold improvement in aqueous solubility over compound **2**) also had poor aqueous solubility, but compared with **2**, **9a** has an amino group in the *para* position of the phenyl B ring that is easier to modify; therefore, we attempted to modify the structure of **9a** to obtain water-soluble compounds. Lipinski proposed the theory of the “Rule of 5” in 1997 [19], which is used to evaluate the pharmacokinetic characteristics of small molecule drugs. Compounds conforming to Lipinski’s rule will have better pharmacokinetic properties and higher bioavailability in the process of metabolism in vivo, so such compounds are more likely to become oral drugs than compounds that do not conform to the rule. Initially, we designed compounds in strict accordance with the “Rule of 5” (Table 1), and the compounds were verified to be in accordance with the Rule of 5 using the online Molinspiration property program [20].

Herein, based on previous research in our laboratory and the availability of the more easily modified **9a**, we designed and synthesised a series of compounds to improve the solubility of stilbene nitriles by introducing water-soluble groups (Figure 1) [21,22,23,24,25]. Specifically, we substituted the amino group of the phenyl B ring with a hydrolysable amide structure and then attached different nitrogen-containing structures. The compounds were further optimised according to the pharmacological and aqueous solubility results, focusing on the effect of the piperazine ring on the aqueous solubility of the compounds. In addition, we investigated the effects of the destruction of the molecular planarity [26,27] and the methoxy group of the benzene A ring on the solubility of the compounds. The molecular docking of the compounds with the enzyme and the drug-like properties of the compounds are also reported herein.

## 2. Results and Discussion

### 2.1. Chemistry

A series of compounds **9a**–**d** were prepared following a six to eight-step synthetic route (Figure 1). Briefly, 3,4,5-trimethoxybenzoic acid was esterified to give **4**. Subsequently, a reduction reaction with LiAlH_4_ was performed at the ester group position, where the resulting intermediate **5** was brominated with phosphorus tribromide solution to give intermediate **6**. After TMSCN (trimethylsilyl cyanide) substitution of the bromine, the resulting **7** was condensed with *p*-nitrobenzaldehyde or 2-methyl-4-nitrobenzaldehyde to afford series **8**. A reduction reaction was then carried out with the SnCl_2_ to afford compounds **9a**–**d**. The resulting **9a** was subjected to amidation with a chloroacetyl chloride to give the amide **10a**. Finally, the halogen moiety was substituted with nitrogen-containing compounds to give compound **11a**, which was soured with hydrochloric acid to give target compounds **12a** and **13**–**20** (Figure 2). The synthesis of **12b**–**12d** (Table 2) was conducted in a manner similar to the preparation of **12a**.

All intermediates and target compounds were purified using column chromatography or recrystallization and characterised by various spectroscopic and analytical methods.

### 2.2. Biological Evaluation

#### 2.2.1. Evaluation of the Cytotoxicity and Aqueous Solubility In Vitro

Initially, we designed and synthesised seven compounds—**9a** and **12a_1_**–**12a_6_**—and evaluated the anticancer activity and aqueous solubility. We evaluated the cytotoxic effects of these stilbene nitrile analogues against a panel of four human cancer cell lines (HCT-116, BEL-7402, MCF-7 and AGS) and two human non-cancerous normal cell lines (L-02 and MCF-10A), using CA-4 as a control. The compounds were screened for aqueous solubility in 0.01 M PBS using a high throughput UV method [28,29,30]. The results are shown in Table 3. Compared with **9a**, **12a_1_**–**12a_6_** showed reduced activity against cancer cells. The activity of the six different nitrogenous compounds **12a_1_**–**12a_6_** against cancer cells was compared, and it was found that the changes in these compounds had little effect on the rate of growth inhibition of the cancer cells. It is possible that the size of the nitrogen heterocycle had little effect on the activity of these compounds. The compounds containing a morpholine ring were generally slightly less active against cancer cells than the other nitrogen-containing compounds. The aqueous solubility of all the compounds **12a_1_**–**12a_6_** was higher than that of **9a**, particularly that of **12a_1_** and **12a_2_** containing piperazine rings. The water solubility of **12a_1_** and **12a_2_** was at least 1687 and 2494 times higher than that of **9a**. Of the six compounds with improved aqueous solubility, **12a_1_** had the best activity. These results indicated that the introduction of a piperazine ring gave the best results for increasing the solubility of the stilbene nitrile parent structure compared with the other amine compounds.

Another strategy to improve the aqueous solubility of compounds is to reduce the planarity [27]. Therefore, we introduced a methyl group into the *meta* position of the benzene B ring to give **9b** and the derivatives **12b_1–2_**, which contained a piperazine ring (Table 2; notably, similar to the **12a** series, a series of compounds containing the other four nitrogen-containing groups (morpholine, pyrrolidine, diethylamine and 4-piperidinemethanol) were also synthesised. Still, these compounds had low activity and poor aqueous solubility). By synthesizing **9c** and **9d** and the respective derivatives, **12c_1–2_** and **12d_1–2_** (Table 2), we could also explore the effect of the methoxy group on the solubility of the stilbene nitrile parent. The anticancer activity and aqueous solubility of all the synthesised compounds were determined, as before, to explore the structure–solubility relationship. The results are shown in Table 4.

The results showed that the aqueous solubility of **9b**, **9d**, **12b_1–2_** and **12d_1–2_** decreased when a methyl group was introduced into the *meta* position of the benzene B ring. This was inconsistent with our previous expectations. Compared with **9a**, the aqueous solubility of the methyl-containing **9b** was decreased by nine times, and this effect was more marked in compounds **12b_1–_2** compared with **12a1–2**. In addition, compounds **9b** and **12b_1–2_** showed a slight decrease in the activity against cancer cells, compared with **9a** and **12a_1–2_**. The toxicity of **9b** and **12b_1_** toward two types of normal cells was increased, but the toxicity of **12b_2_** toward normal L-02 cells was reduced, compared with **9a** and **12a_1–2_**. A similar trend for the anticancer activity and toxicity was observed between compounds **9c** and **9d** and the piperazine ring-containing derivatives **12c_1–2_** and **12d_1–2_**. In short, the results in several cells we screened indicated that compounds without a methyl group on the benzene B ring had increased anticancer activity, increased aqueous solubility and lower toxicity compared with compounds with a methyl group.

The effect of the methoxy group of the benzene A ring on the activity and solubility was investigated by comparing **9a** and **9c** with the respective piperazine derivatives **12a_1–2_** and **12c_1–2_**. The activities of **12c_1–2_** against several cancer cells screened were lower than those of **12a_1–2_**, respectively. It was evident that the three methoxy groups on the benzene A ring are an important part of the pharmacophore, which was in agreement with previous studies [3,4,30]. The solubility was also decreased when one of the methoxy groups of the benzene A ring was replaced with hydrogen (Table 3 and Table 4). This result indicated that when the number of methoxy groups on the benzene A ring was decreased, the lipophilicity of the parent stilbene nitrile was enhanced. This enhancement can offset the effect of the piperazine ring on the aqueous solubility of the parent structure. In summary, the trimethoxyphenyl structure of the parent stilbene nitrile is important for the anticancer activity and water solubility of the compound.

Comprehensive analysis of **9** and **12a_1–6_** showed that **12a_1_** was the most promising compound. Although the activity was slightly reduced, the water solubility was greatly improved. To further investigate the solubility of compounds containing a piperazine group, six piperazine derivatives (Table 2, **15**–**20**) with different substituents or configurations were selected for further study. The compounds were synthesised as described in Figure 2, and the anticancer activity in vitro and solubility were characterised (Table 5). We found that the aqueous solubility of **15** and **16** was significantly improved compared with **9a**; the solubility was increased by more than 231 times compared with **9a**.

Interestingly, **17** and **18** had similar anticancer activity, compared with **19** and **20**, respectively, but had increased aqueous solubility. Importantly, racemic **18** was significantly more soluble than **17** with an (*S*)-configuration. In contrast, the solubility of **20** was not changed appreciably compared with that of **19**. We attributed this result to the different crystal forms of **17** and **18** in water, which affected their solubility in water [33].

The toxicity of anti-tumour drugs toward normal tissues is a very important issue in chemotherapy, so we investigated the cytotoxic effects of these derivatives on two kinds of normal human cells (Table 3, Table 4 and Table 5). The results indicated that the reference compound CA-4 was very toxic to L-02 and MCF-10A normal cells, with IC_50_ values of 1.10 and 3.23 µM, respectively, but most of the synthesised compounds showed only slight cytotoxicity toward the two kinds of normal cells (IC_50_ > 15 µM), the IC_50_ value of some of the compounds, such as **12c_1_**, was >100 µM. In addition, the most active compound **9a** was 1000 times more selective for HCT-116 cancer cells over L-02 normal cells.

After analysing the biological activity and aqueous solubility of these compounds, we further investigated the drug-like properties of **9a** and **12a_1_**, which were the most promising lead compounds.

#### 2.2.2. Determination of the Drug-Plasma Protein Binding Rate

Most drugs are delivered to disease targets through the blood after entering the human body. However, when drugs are mixed with blood, the drugs may combine with many components in the blood, including cells and proteins. Plasma proteins can absorb a large proportion of drug molecules. The binding of a drug with plasma protein will retain the drug in the plasma, limit the distribution of the drug to the diseased tissue, and reduce the metabolism of the drug. A higher dose of the drug will be required to achieve the therapeutic concentration, which increases the potential for toxicity and side effects.

We selected compound **12a_1_**, which had excellent water solubility and activity, and determined the plasma protein binding rate with rat plasma; the results are summarised in Table 6. Compared with the 72.3% binding rate demonstrated for CA-4 in rat plasma [34], a relatively low protein binding rate (60.7%) was observed for **12a_1_** in rat plasma, which may be related to the low lipophilicity of **12a_1_**. However, it should be recognised that this may also be caused by objective differences in experimental conditions, but in general, **12a_1_** maintains at least the same level of plasma protein binding rate as CA-4.

#### 2.2.3. Plasma Stability

Blood contains a large number of hydrolases, and hydrolysis in plasma is a major cause of compound clearance, which can result in pharmacologically efficacious concentrations being unable to be achieved in vivo. Therefore, in vitro determination of the plasma stability of compounds can not only improve the screening of compounds to enable the better selection of lead compounds but can also guide the subsequent modification of selected compounds. The plasma stability of **12a_1_** was investigated because it contains an amide group susceptible to plasma degradation. Compound **9a** is the lead compound of this series of compounds, and it had the best activity against cancer cells, so we tested its stability in plasma as a control.

Compounds **9a** and **12a_1_** were incubated in rat plasma and analysed via HPLC with UV detection at 294 nm (Figure 2). Within 36 h, the peak areas of **9a** and **12a_1_** decreased by approximately 33% and 22%, respectively. However, no additional peaks because of the hydrolysis of these two compounds were detected (Figure 3). We believe that this result may be because (1) the degradation products lacked UV absorption at 294 nm or were not retained on the column, or (2) compounds **9a** and **12a_1_** are, in fact, stable in blood plasma and the decrease in the peak areas was because of the adsorption of protein in the injection bottle or the plasma. In addition, the decrease in the peak area mainly occurred after 24 h, and almost no change occurred before then. Therefore, we tend to believe that the decrease was because of hydrolysis rather than adsorption; that is, owing to a lack of UV absorption and retention, the hydrolysates of **9a** and **12a_1_** were not detected. In either case, the peak areas of the two compounds were almost unchanged within 24 h (Figure 2). Thus, we concluded that these two compounds were stable in the plasma for 24 h.

#### 2.2.4. Molecular Modelling

To further understand the SAR observed in the cell assays, **9a**, **12a_1_** and **12c_1_** were selected for molecular docking studies using *Discovery Studio 3.2* (PDB: 1SA0). In the two-dimensional model shown in Figure 4, **9a** was able to establish hydrogen bonds with more amino acids than CA-4 could, including hydrogen bonds with tubulin. The H and O atoms of the 3,4,5-trimethoxyphenyl group of **9a** were also able to form interactions with Ser-178, Lys-352, Thr-179 and Thr-353. In addition, the cyano group of **9a** formed hydrogen bonds with Lys-352 and Ala-250 (Table 7). Notably, Lys-352, Thr-179 and Ala-250 are vital in the composition of the colchicine binding site, as well as in the formation of hydrogen bonds [35]. In contrast to CA-4, which only acts on β-tubulin, **9a** can interact with both α- and β-tubulin in a more compact way (Figure 5). The docking results for compound **12a_1_** showed that the compound formed more hydrogen bonding interactions, compared with **9a** (Table 7), but lacked interactions with important amino acids, such as Lys-352 and Ala-250, which may be the reason for its reduced anticancer activity. However, **12a_1_**, similar to **9a**, could interact with both α- and β-tubulin, which increased its selectivity index compared with CA-4 [17]. Compared with **12a_1_**, **12c1** with two methoxy groups on the benzene A ring also lacked the hydrogen bond interactions with important amino acids (Figure 4), and the number of hydrogen bonds was appreciably less than for **12a_1_** (Table 7), which may lead to the observed decrease in the anticancer activity. In general, molecular docking studies could qualitatively explain the SAR observed in the biological assays.

## 3. Experimental Section

### 3.1. Chemistry

All the chemicals used in this study were obtained commercially and used were of analytical grade, and THF was dehydrated by 4Å molecular sieve. Melting points were determined with the SGWX-4 melting-point measurement apparatus and were uncorrected. The NMR spectra were recorded on a Bruker AV-300 spectrometer in DMSO-*d6* using tetramethylsilane as the internal standard; chemical shifts were recorded as δ (ppm). High-resolution mass spectra were obtained using MALDI-TOF/TOF mass spectrometer (Bruker Dartonik, Bremen, Germany). Flash chromatography was carried out on silica gel (200–300 mesh), and chromatographic solvent proportions are expressed on a volume: volume basis.

### 3.2. Chemical Synthesis

#### 3.2.1. General Procedure for Synthesis of Compounds **9a**–**d** (Figure 1)

Take **9a**, for example; intermediate **7a** is synthesised according to the methods in the literature [36,37,38,39,40]. Add 5% NaOH aqueous solution (0.20 mL) to the solution of **7a** (1.00mmol) and p-nitrobenzaldehyde (1.10 mmol) in EtOH (10 mL), and the reaction mixture was allowed to stir briskly for 30 min at RT. Acidification of the reaction mixture using 1N HCl resulted in the formation of compound **8a** as a solid, which was recrystallized from EtOH. And then, to a flask was added **8a** (1.00 mmol), ethanol (10 mL) and SnCl_2_ (4.50 mmol) and the mixture was refluxed for about 0.5 h. Neutralized with saturated NaHCO_3_ to weak basicity, diluted with water (about 10 mL), and extracted with dichloromethane. The combined extracted organic phase was dried over anhydrous Na_2_SO_4_, filtered, and evaporated to dry by rotary evaporator to give a yellow solid. To a solution of the obtained yellow solid in THF (10 mL) was added concentrated HCl (1.10 eq), and the reaction mixture was allowed to stir briskly for 8 h at RT. After suction filtration, the filter cake was washed with THF and petroleum ether and dried to give a yellow solid **9a** (0.19 g, 55%). The synthesis of compounds **9b** and **9d** is the same as the above method, except that *p*-nitrobenzaldehyde is replaced by 2-methyl-4-nitrobenzaldehyde (Appendix A), which was obtained by method [41].

#### 3.2.2. General Procedure for Synthesis of Compounds **12a**–**d**, **13**–**18**

Take **12a_1_**, for example; compound **12a_1_** is synthesised according to the methods in the literature [42,43]. To a solution of **9a** (2 mmol) and triethanolamine (2.40 mmol) in THF (10 mL) was added chloroacetyl chloride (2.40 mmol) in THF (3 mL). The mixture was stirred at 0 ℃ for 2–3 h (TLC monitoring response), partitioned between dichloromethane and water; the organic fraction was dried, the solvent removed under reduced pressure to obtain the crude product **10a**. Without purification, the next step can be carried out directly. A total of 1 mmol of intermediate **10a** was dissolved in 10 mL of anhydrous ethanol. Then 1.20 mmol of dry K_2_CO_3_ and a catalytic amount of KI were added, and the reaction mixture was stirred for about 0.5 h at 80 °C. After this time, 1.20 mmol *N*-methyl piperazine was added, and the reaction mixture was stirred for about 3 h at 80 °C (TLC monitoring response), inorganic ingredients were filtered off, and anhydrous ethanol was distilled off on a rotary evaporator. The crude product **11a_1_** was obtained as colored oils. To a solution of the obtained **11a_1_** in THF (5 mL) was added concentrated HCl (1.10 eq), and the reaction mixture was allowed to stir briskly for 8 h at RT. After suction filtration, the filter cake was washed with THF and petroleum ether and dried to give a hydrochloride salt **12a_1_** (0.28 g, 72%).

### 3.3. Solubility and Biological Evaluation

#### 3.3.1. Solubility Assessment

The aqueous solubility was determined as a previously reported procedure [28,29,30,31,32]. Put simply, a certain amount of powdered compound was placed in a 2 mL centrifuge tube and mixed with an appropriate volume of aqueous phosphate buffer (10 mM, pH 7.4) to make the whole system supersaturated. The mixtures were sonicated (KQ5200DE numerical control ultrasonic cleaner, Kunshan Ultrasonic Instrument Co., Ltd., SuZhou, China) for 20 min. After sonication, the mixture was allowed to equilibrate at RT (22 ± 2 °C) for 1.5 h with vortex shaking (1200 rpm). Then centrifuged at 3000 rpm for 15 min (1-14k, sigma, Darmstadt, Germany), the supernatant was filtered with a 0.22 µM filter membrane. And then, take 150 µL of filtrate and add it to 96-well UV plate with triplicate. Draw 150 μL of blank solution was added into plate as blank control. The 96-well UV plate prepared above was read on a Synergy HTX Multifunctional Microplate Reader (BioTek, Vermont, USA) at 294 nm. The absorbance of the sample was corrected according to the UV response of the sample and blank solution, and the corrected UV response values were used for solubility calculations by bringing into the standard curve of the compound.

#### 3.3.2. Materials

3-(4,5-Dimethylthiazol-2-yl)-2,5-diphenyl-2*H*-tetrazolium bromide (MTT) was purchased from Sigma-Aldrich Co. (St. Louis, MO, USA).

#### 3.3.3. Cell Lines and Cell Culture

All human cell lines were used in this study. AGS gastric cancer cell, BEL-7402 liver cancer cell, HCT-116 colorectal cancer cell, L-02 normal liver cell, MCF-7 breast cancer cell, and MCF-10A normal breast cell were initially purchased from American Type Culture Collection (ATCC, Manassas, VA, USA). RPMI-1640 media, DMEM and FBS, were provided by Gibco Company. The cells were maintained in DMEM or RPMI-1640, supplemented with 10% FBS, 100 IU/mL penicillin and 100 mg/mL streptomycin, and at 37 °C in a humidified atmosphere containing 5% CO_2_.

#### 3.3.4. Cell Growth Inhibition Assay (MTT Assay)

The experiment was carried out according to the method of reference [44].

#### 3.3.5. Study of Compound Stability in Plasma

##### Plasma Preparation

Blood was collected from SD rats (200–250 g, male and female, Animal Experimental Center of Yanbian University) into heparin tubes, and supernatant plasma was collected after centrifugation.

##### Study of Compound Stability in Plasma

A certain amount of **9a** and **12a_1_** were accurately weighed and mixed with rat plasma. After incubation at 37 ℃ for 0, 6, 12, 24 and 36 h, the plasma samples were treated with methanol and analyzed by HPLC to determine their stability in plasma.

Analytical instrument and conditions:

A HITACHI Primaide HPLC liquid chromatography system was equipped and used for all analyses. Chromatographic separation was performed on a Welch Ultimate XB-C18 column (3.9 mm × 150 mm, 5 μm; Welch, China). The analytes were eluted using a gradient method with a flow rate of 1 mL/min. Mobile phase A was Ultra pure water, and mobile phase B was methanol according to the following gradient: (1) The gradient elution conditions of **9a**: 0 to 8 min, 55% to 67% B; 8.1 to 15 min, 67% to 90% B; 15.1 to 25 min, 55% B. (2) The gradient elution conditions of **12a_1_**: 0 to 9 min, 55% to 65% B; 9.1 to 12 min, 65% to 75% B; 12.1 to 15 min, 75% to 85% B; 15.1 to 20 min, 90% B; 20.1 to 30 min, 55% B. The injection volume was 3 μL. The column temperature was set at 40 °C.

#### 3.3.6. Drug-Plasma Protein Binding Rate Determination

Compounds at 90 μM concentration were mixed with rat plasma, and the mixtures were subjected to equilibrium dialysis versus phosphate buffer (pH 7.4) at 4 °C for 48 h. The dialysis membrane of molecular weight cut off 1 kDa was used. Dialysis experiments were done in triplicate. On completion of the dialysis period, absorb a certain amount of buffer and plasma, respectively, and add methanol to treat the sample. Dialysis samples were analysed by HPLC. The plasma protein binding rate of the compound was calculated as follows: protein binding rate/% = (plasma drug concentration inside dialysis bag—dialysate drug concentration outside dialysis bag)/plasma drug concentration inside dialysis bag × 100%. The analytical instrument and conditions are shown in Figure 2.

## 4. Conclusions

Our previous studies focused on the optimisation of the activity of CA-4 derivatives against cancer cells. They led to the identification of **2**, a potent, stable and selective anticancer compound but with poor aqueous solubility. The work presented herein describes several strategies that were employed to improve the solubility of CA-4 derivatives. To study the effects of different modifications on the aqueous solubility of the compounds, we designed and synthesised a series of CA-4 derivatives containing different molecular cores, divided into four series: **12a**, **12b**, **12c** and **12d**. The most successful approach was to incorporate amides containing an aqueous because of the nature of the parent group at the *para* position of the benzene B ring (the **12a** series). The solubility of **12a_1_**, containing methylpiperazine, was 1688-fold higher than that of **9a**, but this modification led to a decrease in the anticancer activity compared with the parent compound. 

Further exploration of the **12a_1_** chemotype led to the synthesis of **15–20**. Compared with **9a**, the anticancer activity was decreased, and the solubility of **15** and **16** was improved more than 231-fold. Interestingly, biological evaluation of the compounds revealed that intermediate **9a** had the highest anticancer activity. However, the aqueous solubility was five times lower than that of CA-4, and the activity against HCT-116 was eight times higher than that of CA-4, reaching sub-micromolar levels (IC_50_ = 0.02 µM). There was no apparent relationship between the water solubility and the anticancer activity of the compounds, which was consistent with the results published by Álvarez [45].

In terms of the development of active compounds in plasma, **9a** and **12a_1_** retained plasma stability over 24 h, and the drug-plasma protein binding rate of **12a_1_** was maintained at a medium level. The results of molecular docking studies showed that **9a** and **12a_1_** could bind with key amino acids at the colchicine binding site of tubulin via hydrogen bonds.

This research aimed to improve the water solubility of CA-4 derivatives; our research greatly improved the water solubility of such structures on the premise of maintaining high pharmacological activity. This study laid a good foundation for further promoting the drug formation of such structures. However, many related studies on the drug formation of such structures still need to be continued, such as more comprehensive testing of the pharmacokinetic characteristics of compounds, especially in animals.

## Data Availability

Not applicable.

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
