# Peer review of "Development of Combretastatin A-4 Analogues as Potential Anticancer Agents with Improved Aqueous Solubility"

_molecules, 2023, doi:10.3390/molecules28041717_

Round 1

Reviewer 1 Report

1. Clarify the sentence in the Abstract"A kinetic solu-12 bility assay revealed that the solubility of compounds containing a piperazine ring increased the 13 most,"

2. Add the cytotoxicity data of the most active compound in the Abstract.

3. Add a few sentence about the molecular target (tubulin) in the introduction section.

4. Some typographical error was found in Line 104, and 105.

5. In the molecular docking studies, represent the docking score and types of binding interaction with amino acid residue in Tabular form to easily understand the mode of interaction.

6. Over all add spectra (1H NMR, 13C NMR and HRMS) of some of the important synthetic compounds in the Supplementary Information.

Author Response

  1. Clarify the sentence in the Abstract"A kinetic solu-12 bility assay revealed that the solubility of compounds containing a piperazine ring increased the 13 most,"

【Answer】The sentence has been confirmed and revised.

  1. Add the cytotoxicity data of the most active compound in the Abstract.

【Answer】The data has been added in the Abstract.

  1. Add a few sentence about the molecular target (tubulin) in the introduction section.

【Answer】It has been added in the introduction section.

  1. Some typographical error was found in Line 104, and 105.

【Answer】The error has been revised.

  1. In the molecular docking studies, represent the docking score and types of binding interaction with amino acid residue in Tabular form to easily understand the mode of interaction.

【Answer】Thank for you suggestion. A new table (Table 7) related the request has been uploaded.

  1. Over all add spectra (1H NMR, 13C NMR and HRMS) of some of the important synthetic compounds in the Supplementary Information.

【Answer】Thank you. The new “Supplementary Information” contains spectra (1H NMR, 13C NMR and HRMS) of all of the synthetic compounds.

Reviewer 2 Report

This manuscript describes several structural modifications of combretastatin A-4 to improve its solubility of CA-4. A kinetic solubility assay revealed that the solubility of compounds containing a piperazine ring increased the solubility. The work has been carried out with care and the results have been presented with clarity and discussed appropriately. In addition, these synthesized stilbene nitriles had high anticancer selectivity, while maintaining micromolar activity against cancer cells. Preliminary studies indicated that compound 12a1 had excellent plasma stability and moderate binding to rat plasma proteins, suggesting it is a promising lead compound for the development of an anticancer agent. In this way, this manuscript demonstrates a significant contribution in the area of medicinal chemistry. Accordingly, I can recommend publication of this work in Molecules after solving the issues detected.

Issues:

- The authors should check formatting errors in the reference citations.

- The authors should check the format of Figure 3.

- The authors should check the format of the bibliography.

Author Response

  1. The authors should check formatting errors in the reference citations.

【Answer】We checked carefully in formatting errors in the reference citations and have been revised.

  1. The authors should check the format of Figure 3.

【Answer】It has been revised.

  1. The authors should check the format of the bibliography.

【Answer】We checked it and revised.

Reviewer 3 Report

Tian and coworkers reported a recent development of new potential anticancer agents after a deeply investigation of the aqueous solubility of these new compounds.

The authors discuss about the synthesis of several Combrestatin A-4 analogues and how different chemical parameters can affect the stability and solubility of these molecules.

This review should be accepted after the authors have addressed some issues.

Minor revisions:

Please check carefully in all the manuscript the chemical structures in order to obtain the same dimension of all the molecules, in all the figures (for example Scheme 2 is completely different from Scheme 1).

Please check carefully in all the manuscript formatting of the references because a lot of times are uncorrected (see for example ref. 6 and 7).

Page 2. line 79, please provide at least the reference of the previous structure-activity relationship studies or introduce some examples of these studies.

Please change ice with 0°C in all the text.

Please in the experimental section underline in bold all the compound synthetized in this paper.

For the Supporting Information please introduce a Table of content with different paragraphs in order to clarify the reporting data

Please provide all the spectra for the new compounds synthetized and all the mass spectra in the supporting information.

Author Response

Minor revisions:

1. Please check carefully in all the manuscript formatting of the references because a lot of times are uncorrected (see for example ref. 6 and 7).

【Answer】 We checked carefully in all references and have been revised. For example, Ref.6 and 7 have been revised.

2. Page 2. line 79, please provide at least the reference of the previous structure-activity relationship studies or introduce some examples of these studies.

【Answer】 The related reference has been marked in the manuscript.

3. Please change ice with 0°C in all the text.

【Answer】The “ice” in all synthetic conditions of the manuscript was changed by “0°C”.

4. Please in the experimental section underline in bold all the compound synthetized in this paper.

【Answer】The compounds synthesized in this paper are underlined in bold.

5. For the Supporting Information please introduce a Table of content with different paragraphs in order to clarify the reporting data.Please provide all the spectra for the new compounds synthesized and all the mass spectra in the supporting information.

【Answer】Thank you very much for your suggestion. According to the requirements, a new Supporting Information file was prepared. It provide the detailed contents, such as “Table of contents”, all the spectra for the new compounds synthesized and all the mass spectra.

Round 2

Reviewer 1 Report

Manuscript is ok now.